

# A wavelet-based-approach to detect climate change on the coherent and turbulent component of the atmospheric circulation

Davide Faranda[1] and Dimitri Defrance[1]

[1] LSCE-IPSL, CEA Saclay l'Orme des Merisiers, CNRS UMR 8212 CEA-CNRS-UVSQ, Université Paris-Saclay, 91191 Gif-sur-Yvette, France

*Correspondence to:* Davide Faranda (davide.faranda@lsce.ipsl.fr)

**Abstract.** The modifications of atmospheric circulation induced by anthropogenic effects are difficult to capture because wind fields feature a complex spectrum where the signal of large scale coherent structures (planetary, baroclinic waves and other long-term oscillations) is mixed up with turbulence. Our purpose is to study separately the effects of climate changes on these two components by applying a wavelet analysis to the 700 hPa wind fields obtained in climate simulations for different forcing scenarios. We study the coherent component of the signal via a correlation analysis to detect the persistence of large-scale or long-lasting structures, whereas we use the theory of Auto-Regressive Moving-Average stochastic processes to measure the spectral complexity of the turbulent component. Under strong anthropogenic forcing, we detect a significant climate-change signal. The analysis suggests that coherent structures will play a dominant role in future climate, whereas turbulent spectra will approach a classical Kolmogorov behavior.

## 1 Introduction

Scale separation is an essential property for the study of natural systems: Langrangian mechanics has been applied to the study of the solar system because planets appear so small that they can be considered as material points with respect to the length of their orbits (Murray and Dermott, 1999). In a less obvious framework, Einstein and Langevin recognized that the behavior of heavy particles in a gas can be studied by introducing two different scales: the inertial (slow) motion of the heavy particles and the interactions (fast) with the gas particles (Langevin, 1908). In geophysics the same approach has been - sometimes implicitly - applied for understanding important mechanisms driving the atmospheric and oceanic circulation: one can model the baroclinic instability because cyclones have a well determined size and their structure emerges out from the atmospheric turbulence (Charney, 1947), El Nino because of the precise time-scales involved in the phenomenon (Cane and Zebiak, 1985; Penland and Magorian, 1993). The success of a scale-separation based approach is due to the intrinsic properties of stratified and rotating flows. In homogeneous and isotropic turbulence, the energy flows towards the small scales and coherent structures are rapidly destroyed. This is the so-called direct cascade proposed in Kolmogorov (1941)). Instead, geophysical flows are stratified and rotational flows where an inverse cascade of energy induces the bidimensionalization of motions and contributes to the formation of large scale coherent structures (Pouquet and Marino, 2013). In laboratory experiments (Lamriben et al., 2011) and geophysical observations (Craig and Banner, 1994; Holmes et al., 1998) one aims at separating coherent and turbu-





lent components and build theoretical models to describe the associated motions (Charney, 1971; Pitcher, 1977; Jiang et al., 1995; Lucarini et al., 2007). As pointed out by several authors (see Schertzer and Lovejoy (1991) for a review), this task is non-trivial because both the inverse and direct cascades coexist for geophysical motions. The direct cascade is eventually responsible for the dissipation of energy, the transfer of momentum from the atmosphere to the ocean and the soil, the disruption

of large scale structures in the flow resulting in an unpredictable behavior (Leith, 1971). The inverse cascade contributes to the formation of cyclonic and anticyclonic structures observed in the atmosphere and the ocean. Morevoer, it generally enhances the predictability of future states of the atmosphere (Paladin and Vulpiani, 1994; Tribbia and Baumhefner, 2004).

In this paper we present two indicators that describe the statistical properties of large scale coherent structures as well as
turbulent spectra, investigating their response to climate change. The indicators are defined after separating the coherent structures from featureless turbulence via the wavelet filtering technique. For the coherent part, we compute the integral of the auto-correlation function as a measure of the persistence of the coherent structure. For the turbulent component, we use an indicator that measures the complexity of the spectrum with respect to the canonical behavior theorized by Kolmogorov.

We test the technique on the horizontal wind data measured at 700hPa for two different anthropogenic emission scenarios (RCP 2.6 and 8.5). We investigate whether the anthropogenic and natural forcing could cause not only change the intensity of some defined observable (as it is now evident for the global mean temperature), but also in the direction the energy is cascading and therefore in the relative importance of large scale coherent structure with respect to turbulence.

## 2 Methods

The separation between coherent $X(t)$ and turbulent $Y(t)$ components of a time series $Z(t)$ is done via wavelet filters (Farge, 1992). With respect to simple filtering technique (e.g. moving-average filters), the wavelet filters are useful when the time series contain multiple timescales and there is not a trivial scale separation.

We give a brief illustration of the wavelet filtering technique by analyzing a time series of $u_{500}$ taken from the scenario RCP 2.6 at the point lon 78W lat 38N. The series consisting of 512 monthly observations and it is shown in Fig. 1a). Its power
spectral density (psd) is visualized in Fig. 1b. The series shows an evident periodic component (the seasonal cycle) captured by the spectral peak. The spectrum is compared with a flat one (dotted line) reproducing a perfect white noise signal. The results of the wavelet filter are shown in Fig.1c-f). The coherent component $X(t)$ and its spectrum are shown in Fig.1c,d) respectively. The effects of the filter are not directly visible on the detrended time series, but rather on the spectra. The psd for the coherent part of the signal presents a significant slope with the energy concentrated at large time scales. On the contrary, the incoherent
component $Y(t)$ represented in Fig.1e) has a rather flat psd (Fig.1f) as expected from a successfull application of the technique.





Once the separation between coherent and noisy component is done, we study the property of $X(t)$ and $Y(t)$ separately. For the coherent component $X(t)$ we use as indicator the memory of the system by measuring the integral of the autocorrelation function defined as:

$$ACF(X)(\tau) = E[X(t)X^*(\tau)],$$

where $E[X]$ stands for expectation value. The $ACF$ measures how long the system remember an initial condition. For a white noise signal, it decays to 0 as $\tau > 1$. For a correlated signal it decay slowly to 0 for large $\tau$. For a perfectly periodic signal, the $ACF$ is periodic itself. The integral of the $ACF$ in its discrete version is written us:

$$\Lambda = \sum_{\tau=0}^{T} ACF(X)(\tau),$$

where we sum the correlation up to a time $T$ sufficiently large for the ACF to decay to 0. $\Lambda$ measures how long coherent
structures persist in time and it is therefore linked to the predictability: the higher the correlation, the higher the probability that the structure will be preserved in future times (Schubert et al., 1992).

For the noisy component $Y(t)$ we use an indicator of the spectral complexity with respect to the canonical Kolmogorov behavior. In order to introduce this indicator we will use the class of Auto Regressive Moving Average stochastic processes. In
general, a stationary time series $Y_t$ of an observable with unknown underlying dynamics can be modeled by an ARMA$(p,q)$ process such that for all $t$:

$$Y_t = \sum_{i=1}^{p} \phi_i Y_{t-i} + \varepsilon_t + \sum_{j=1}^{q} \theta_j \varepsilon_{t-j} \tag{1}$$

with $\varepsilon_t \sim WN(0,\sigma^2)$ - where $WN$ stands for white noise - and the polynomials $\phi(z) = 1 - \phi_1 z_{t-1} - \cdots - \phi_p z_{t-p}$ and $\theta(z) = 1 - \theta_1 z_{t-1} - \cdots - \theta_q z_{t-q}$. Notice that, hereinafter, the noise term $\varepsilon_t$ will be assumed to be a white noise, which is a very general
condition (Box and Jenkins, 1970).

The basic model for the noisy component is the ARMA(1,0) or simply AR(1) model which is the simplest compatible with the Kolmogorov spectrum (Thomson, 1987). When the spectral complexity increases, the best ARMA model describing the velocity time series will deviate from the basic one. We can define a normalized distance between the reference ARMA$(\overline{p}, \overline{q})$ and any other ARMA$(p, q)$ model as the normalized difference between the $BIC(n,\hat{\sigma}^2,p+1,q)$ and the ARMA$(\overline{p}, \overline{q})$
$BIC(n,\hat{\sigma}^2,\overline{p}, \overline{q})$:

$$\Upsilon = 1 - \exp\left\{|BIC(p+1,q) - BIC(\overline{p},\overline{q})|\right\}/n. \tag{2}$$

with   $0 \leq \Upsilon \leq 1$: it goes to zero if the dataset is well described by an ARMA$(\overline{p}, \overline{q})$ model and tends to one in the opposite case.

We have already checked that such indicators perform well in different physical systems, providing more information than
the usual ones, based on the critical slow down due to the increase of correlations in the systems at the transition. These analyses




have been recently published in (Faranda et al., 2014) where indicators similar to $\Upsilon$ have been used to model different physical systems: Ising and Langevin models and turbulence. A large $\Upsilon$ correspond to a complex spectrum with non-trivial scale-interactions and non-constant energy transfers, a small $\Upsilon$ correspond to a spectrum compatible with the Kolmogorov spectrum with constant energy fluxes between scales. The predictability will decrease with an higher $\Upsilon$ because more structures at
different scales will have to be followed to describe the behavior of the system. In other words, for high $\Upsilon$ the component $Y(t)$ cannot be just modelled as simple noise.

## 3  Analysis

We illustrate the potential of $\Lambda$ and $\Upsilon$ indicators on a climate change experiment used in the CMIP5 framework for the IPCC AR5 report (Collins et al., 2013). To explore the climate change of the next century, the IPCC has developed four different sce-
narios, defined in terms of radiative evolution and corresponding to a concentration of greenhouse gases year by year between 2006 and 2100 and extended until 2300. Here we consider two scenarios: i) the low emission scenario (RCP2.6) leading to a radiative balance of 2.6 W/m2 in 2100 with a peak at 3W/m2 and a decreasing trend ii) the higher emission scenario (RCP8.5) predicting an increase up to 8.5W/m2 in 2100. The effect of such greenhouse gases perturbations are well known for some observables, e.g. the global temperature increase ranges from 1±0.4 °C to 3.7±0.7 °C in the last part of the period 2081-2100
(Collins et al., 2013).

We focus on the daily horizontal winds at 700 hPa $(u_{700}, v_{700})$ obtained from the IPSLCM5-LR model. This model is developed by the Institut Pierre-Simon Laplace with several laboratories. It consists of several components : atmosphere (LMDZ), ocean (NEMO), continent (ORCHIDEE) and sea-ice (LIM). For LMDZ, a spatially resolution is of 3.75 X 1.875 in longitude
and latitude respectively with 39 vertically levels and, for NEMO, a spatially resolution of about $2°$, with a higher latitudinal resolution of $0,5°$ in the equatorial ocean, and 31 vertically levels. ORCHIDEE takes into account the evolution of the lands (urbanization, forests, agriculture) (Dufresne et al., 2013). We chose the 700 hPa (about 3km) because it has been recognized as the best level for tracking the coherent atmospheric structures as shortwaves, extra-tropical cyclones and convective storms, since they are all advected, in first approximation, by 700 hPa horizontal winds (Mölders and Kramm, 2014).


We begin the analysis by showing typical maps of $\Lambda$ and $\Upsilon$ for the scenario RCP 2.6 and the two components: the zonal $u_{700}$ and the meridional $v_{700}$ (Fig. 2). $\Lambda$ shows, for the zonal component, a rich structure with several areas where persistent coherent structures are well identified (Fig. 2a). Because we are using daily time series, the values of $\Lambda$ can be directly interpreted as the number of days of persistence of these structures. At the mid-latitudes, the signature of stationary planetary waves is
visible. In correspondence to the location of such waves, we find $\Lambda \sim 16$ days, a value compatible with the work of Torrence and Compo (1998) who studied planetary waves using other indicators based on the wavelet approach. At the tropics, the high values of $\Lambda$ can be linked to the easterly jets (Koteswaram, 1958). The core of the African easterly jet is located about at this level (Nicholson, 2009), as well as for the Choco and Caribbean low level jets (Wang, 2007). The results for $\Lambda$ computed on



the meridional component $v_{700}$ are shown in Fig. 2c). Here the largest values are found in correspondence of the regions affected by monsoons. The strongest signal is for the African monsoon because the IPSL model localizes it better than the Indian one (Dufresne et al., 2013). At the mid-latitudes the patches visible near the Pacific coast and over USA correspond to areas where the zonal flow is blocked by the Rocky Mountains and meridional winds blow to allow the flows go round the mountains.

For the $\Upsilon$ analysis, there is not much difference ibn the structure of the zonal component (Fig. 2b) and the meridional component (Fig. 2d). The spatial pattern of $\Upsilon$ can be explained in light of the tropical atmospheric dynamics. Higher values are located on the tropics, where convective storms are the major actor in determining the weather. The non-trivial interplay between deep convection and other meteorological turbulent phenomena affecting the tropics is responsible for high $\Upsilon$ values.

We now investigate whether $\Lambda$ and $\Upsilon$ can detect changes in the coherent or noisy components of the 700 hPa horizontal winds under the climate change RCP 2.6 and RCP 8.5 scenarios. We divide the daily time series in two period: 2005-2055 and 2055-2105 and compute the quantities: $\Delta\Lambda = \Lambda_{2055-2105} - \Lambda_{2005-2055}$ and $\Delta\Upsilon = \Upsilon_{2055-2105} - \Upsilon_{2005-2055}$. For both the indicators, the RCP 8.5 scenario shows considerable impacts, whereas the changes for the RCP 2.6 are appreciable only for

the $u_{700}$ component.

The $\Delta\Lambda$ fields for the $u_{700}$ and the $v_{700}$ components present interesting structures: the strongest signals are located over the Pacific ocean in correspondence of the El Nino Region 3 (for $u_{700}$) and Region 3.4 for $v_{700}$, described in Trenberth (1997). At the midlatitudes, dipolar structures appear both for the zonal and for the meridional flow (Barnston et al., 1997). This is the

signature of the jet stream shift observed by several studies. For the northern hemisphere, changes in the pattern of meridional winds are highlited by an alternance of negative and positive $\Delta\Lambda$: we can link these changes to the modification in stationary planetary waves associated to the change in the jet streams intensity and positions. Several studies have recently appeared on this issue, although is not clear whether the cause can be linked to the so-called Artic amplification (Serreze and Francis, 2006), rather than to changes in El Nino Southern Oscillation (Moritz et al., 2002), or even to the stratospheric dynamics (Serreze

and Barry, 2011). $\Delta\Lambda$ also suggests that some regions will experience an increasing persistence of meridional winds i.e. an increasing of blocking conditions (e.g. central USA, western Europe), whereas some other areas will have an increase in the persistence of the zonal winds (e.g; Eastern Europe, Alberta (CA)).

The results for $\Delta\Upsilon$ indicates that the spectral complexity in the tropical regions tends to reduce of about 10% ($\Delta\Upsilon \sim -0.1$)

in the RCP 8.5 scenario. A possible explanation for this result relies on the enhanced convective activity resulting from the sea surface warming and resulting in an increase of precipitations in the area, as we have verified for the IPSL model and reported by other studies (Huang et al., 2013). If turbulence becomes stronger and anisotropic, then it should approach the Kolmogov behavior, and $\Upsilon$ should tend to 0.



## 4 Conclusions

We have devised two indicators to study the changes in the atmospheric circulation by separating the coherent structures from the turbulent part of the signals. The indicator for the coherent structures $\Delta\Lambda$ is a measure of the total persistence of the coherent structure, whereas $\Delta\Upsilon$ is a measure of the residual complexity of the turbulent spectrum, once the coherent component has been removed.

The indicators show significant changes when the climate system is subject to greenhouse gases forcing. The difference in the indicators for the RCP 8.5 scenario between the second half and the first half of the 21st century suggests that El Nino Southern Oscillation will play a major role and blocking conditions will change the typical coherent structures observed at the mid-latitude.

Besides the regional patterns, we believe that the most important message is contained in the global average of our indicators. For the RCP 8.5 scenario, $\Delta\Lambda$ increases by 0.5 days in the second half of the century for $u_{700}$ and by 0.2 days for $v_{700}$. On the other hand, the spectral complexity decreases in the tropical regions of about 10%. This suggests that the coherent structures will play a major role in the atmospheric dynamics and this will probably enhance the predictability of the atmosphere on weekly to monthly time-scales. The contrast between these two effects could be one of the causes of the difficulty in finding significant traces of climate change in the circulation dynamics, a problem recently highlighted by Shepherd (2014).

Our indicators have been here illustrated for a single model and in two climate change scenarios. They could be useful to evaluate the response of atmospheric circulation to changes in the forcing for several models. Moreover the technique does not require the variables to be velocity fields and it could be extended to any physical time series in which a non-trivial scale separation is present.

*Acknowledgements.* D. Faranda was supported by ERC grant No. 338965-A2C2



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





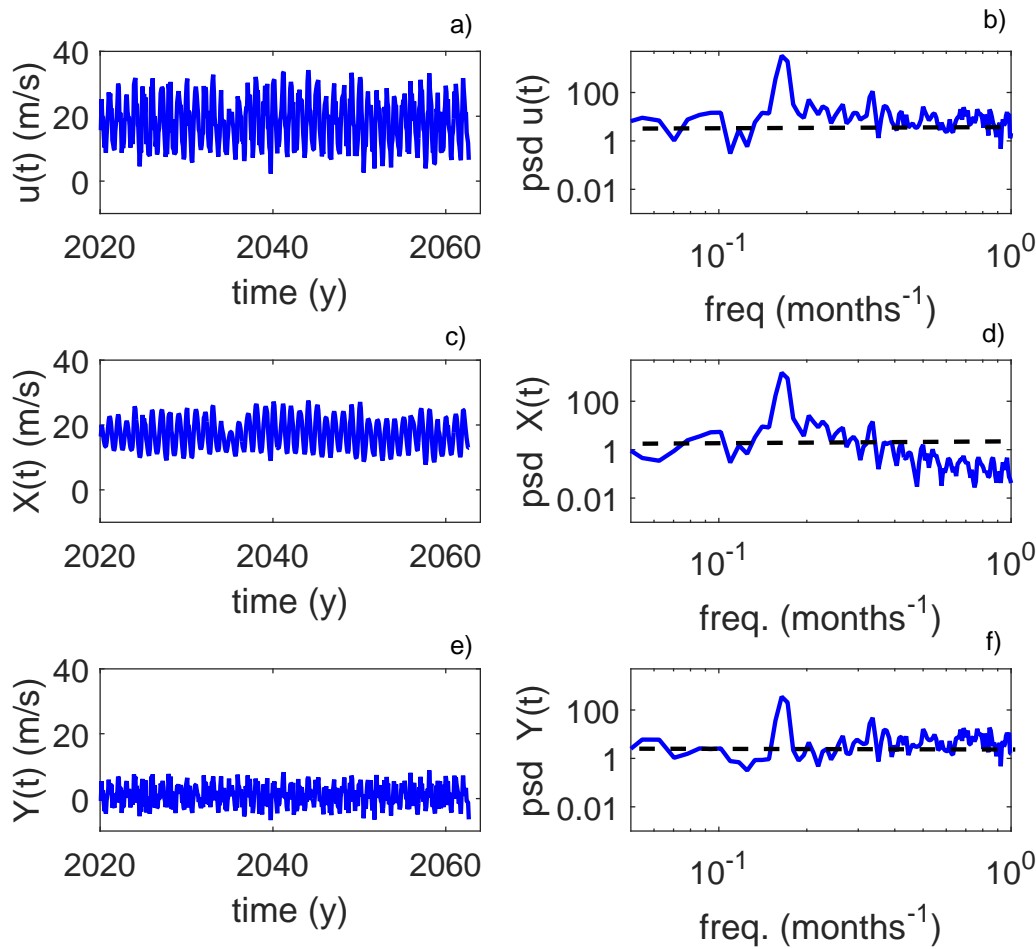

**Figure 1.** Example of wavelet filtering. a) $u_{500}$ monthly time series at Lon=78W and Lat=38N and b) corresponding power spectral density (psd). c) Time series of the coherent component $X(t)$ extracted by the wavelet filter and d) its psd. e) Time series of the noisy component $Y(t)$ extracted by the wavelet filter and f) its psd.





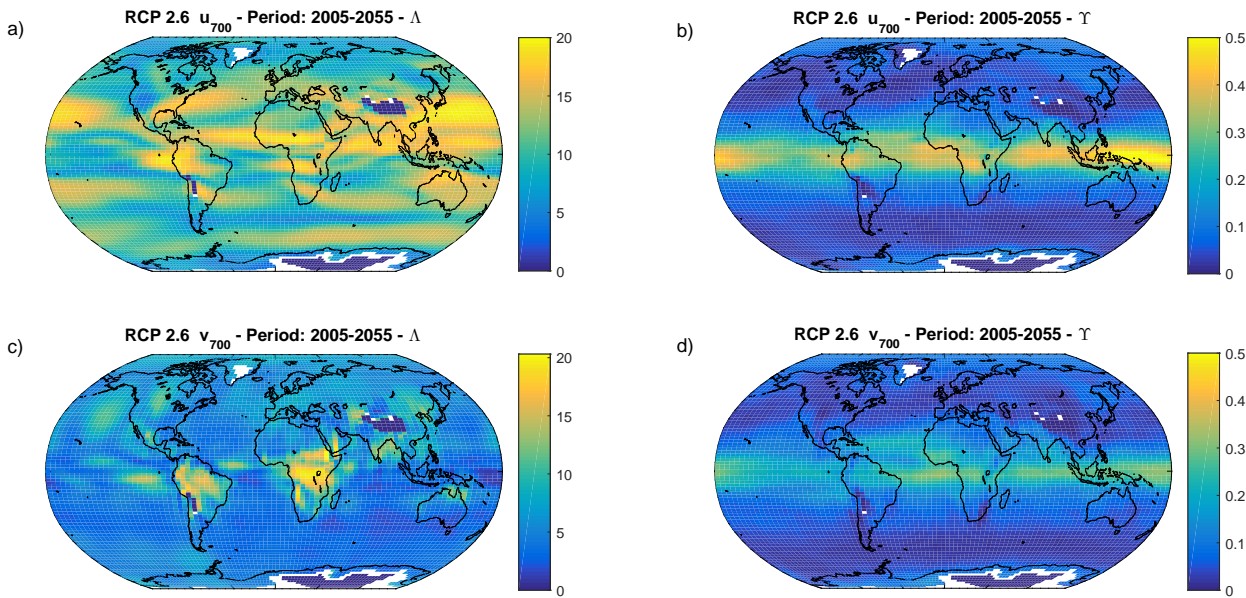

**Figure 2.** Integral of the autocorrelation function $\Lambda$ for the period 2005-2055 for the $u_{700}$ a) and for the $v_{700}$ c) daily time series. Spectral complexity $\Upsilon$ for the period 2005-2055 for the $u_{700}$ b) and for the $v_{700}$ d) daily time series.





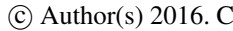

**Figure 3.** Differences in the integral of the autocorrelation function $\Delta\Lambda$ for the 2055-2105 and the 2005-2055 period for the $u_{700}$ in the RCP 2.6 a) and in the RCP 8.5 b). Differences in the Spectral complexity $\Delta\Upsilon$ for the 2055-2105 and the 2005-2055 period for the $u_{700}$ in the RCP 2.6 c) and RCP 8.5 d) scenario. $\Delta\Lambda$ for $v_{700}$ for the RCP 2.6 e) and the RCP 8.5 f) scenarios. $\Delta\Upsilon$ for the $v_{700}$ in the RCP 2.6 g) and RCP 8.5 h) scenario.