# Peer review of "A wavelet-based-approach to detect climate change on the coherent and turbulent component of the atmospheric circulation"

_Earth System Dynamics, 2016_

## Referee Comment (RC1) · Anonymous Referee #1 · 22 Mar 2016

Overall quality
This is a good paper, with some limitations as to the broader conclusions. The authors apply new wavelet based metrics on the changes in circulation under a changing climate and the use of these novel diagnostics is the strongest part of this paper. However, the authors have applied this methodology to a coarse resolution atmospheric simulation that may not be adequate to draw all of the conclusions stated in the paper. This flaw can be rectified, however, by reducing the scope of the conclusions for reasons stated below.

[Figure]

General comments

This paper has three main parts: 1) the use of a wavelet filter to separate the coherent and turbulent components of the 700 hPa wind field; 2) The analysis of these fields using two novel metrics in a simulation of the present day climate; 3) the analysis of the changes in these metrics in future climates simulate using RCP 2.5 and RCP 8.5 warming scenarios.

There are questions with respect to the implementation of the first part and the interpretation of the third part. With respect to the implementation of the first part the, filter and separation, the concern is that the turbulent component has a large seasonal cycle contribution; as large as the coherent part of the flow. This appears to be a an inadequacy of the filter since one would assume that there should be only a small contribution from seasonal fluctuations in the turbulent component. It would be comforting if the authors at least noted this problem in the paper.

Second, the authors interpret the analysis of their results indicating a relative increase in the coherent component versus the turbulent component under the RCP 8.5 scenario as pointing to an increase in predictability. This may be true, however, only for the scales that are well resolved in the IPSLCM5-LR model used; i.e. structures on the synoptic and planetary scales well resolved by a 3.75°x1.875° mesh. This says nothing about what may happen to meso-scale and convective scale phenomena that are unresolvable with such a coarse resolution as used here.

Technical corrections

Pg 2 line 16 ' only change the intensity' should be ' only a change in the intensity'
Pg 4 line 4 'an higher' should be 'a higher'
Pg 4 line 31 'At the tropics' should be 'In the tropics'
Pg 5 line 6 'ibn' should be 'in'

Pg 5 line 21 'hilited by an alternance' should be 'highlighted by an alteration'
Pg 5 line 29 'of about' should be ' by about'

---

## Referee Comment (RC2) · Anonymous Referee #2 · 28 Apr 2016

The wavelet-based approach is up to my knowledge a novel methodology for analyzing climate models, though it has been used for image or reservoir reconstructions. I, however, have big doubts that this method is suitable for climatology. Climate modeling demands multi-scale modeling as well but the scale separation is often difficult to define and what is more important there is a multi-scale interaction that evolves in time. Therefore, the method should be first rigorously examined for climate models (starting from toy models and propagating towards more complex models) before drawing the conclusions about the climate system itself.

[Figure]

Authors claim that the integral of the ACF detects the predictability. However, for that not only the correlation should be high but the error should be small, which is not shown.

Authors test the metrics on one resolution model. However, one needs to show that the wavelet-based separation gives satisfactory results by considering models with different resolutions.

Authors claim that the difference between $\Lambda_{2055-2105}$ and $\Lambda_{2005-2055}$ detects the predictability. I am wondering about sensitivity of this metric with respect to the time interval.

Moreover, authors need to describe the wavelet-based approach, define what BIC is, and to explain how the parameters were chosen.
* * *

---

## Author Comment (AC1) · 23 May 2016

REFEREE: "Overall quality. This is a good paper, with some limitations as to the broader conclusions. The authors apply new wavelet based metrics on the changes in circulation under a changing climate and the use of these novel diagnostics is the strongest part of this paper. However, the authors have applied this methodology to a coarse resolution atmospheric simulation that may not be adequate to draw all of the conclusions stated in the paper. This flaw can be rectified, however, by reducing the scope of the conclusions for reasons stated below. This paper has three main parts:

1) the use of a wavelet filter to separate the coherent and turbulent components of the 700 hPa wind field; 2) The analysis of these fields using two novel metrics in a simulation of the present day climate; 3) the analysis of the changes in these metrics in future climates simulate using RCP 2.5 and RCP 8.5 warming scenarios. There are questions with respect to the implementation of the first part and the interpretation of the third part. With respect to the implementation of the first part the, filter and separation, the concern is that the turbulent component has a large seasonal cycle contribution; as large as the coherent part of the flow. This appears to be an inadequacy of the filter since one would assume that there should be only a small contribution from seasonal fluctuations in the turbulent component. It would be comforting if the authors at least noted this problem in the paper."

**ANSWER:We will comment and discuss this issue in the new version of the manuscript. As visible from figure 1, the seasonal cycle carries indeed a relevant part of the energy spectrum. Our considerations were mostly related to the slope of the spectrum.**

REFEREE: "Second, the authors interpret the analysis of their results indicating a relative increase in the coherent component versus the turbulent component under the RCP 8.5 scenario as pointing to an increase in predictability. This may be true, however, only for the scales that are well resolved in the IPSLCM5-LR model used; i.e. structures on the synoptic and planetary scales well resolved by a 3.75âŮęx1.875âŮę mesh. This says nothing about what may happen to meso-scale and convective scale phenomena that are unresolvable with such a coarse resolution as used here."

**ANSWER: The reviewer is right in saying that we have to focus our conclusions on the synoptic and planetary scales resolved by the model. We will understate the results in the new version of the paper.**

REFEREE: "Technical corrections : Pg 2 line 16 ' only change the intensity' should be ' only a change in the intensity' Pg 4 line 4 'an higher' should be 'a higher' Pg 4 line

31 'At the tropics' should be 'In the tropics' Pg 5 line 6 'ibn' should be 'in' Discussion paper Pg 5 line 21 'hilited by an alternance' should be 'highlighted by an alteration' Pg 5 line 29 'of about' should be ' by about'"

**ANSWER: The typos will be corrected in the new version of the paper**

---

## Author Comment (AC2) · 23 May 2016

*REFEREE:"The wavelet-based approach is up to my knowledge a novel methodology for analyzing climate models, though it has been used for image or reservoir reconstructions. I, however, have big doubts that this method is suitable for climatology. Climate modeling demands multi-scale modeling as well but the scale separation is often difficult to de- fine and what is more important there is a multi-scale interaction that evolves in time. Therefore, the method should be first rigorously examined for climate models (starting from toy models and propagating towards more complex models) be-*

*fore drawing the conclusions about the climate system itself."*

**ANSWER**:The wavelet approach has been devised for analyzing turbulent signals containing non-trivial scale separations. The original paper by Farge (1992) contains applications of wavelet filtering for toy models as well as for turbulent complex systems. This paper has, up to date, about 1500 citations, corresponding to just as many applications in complex fluid mechanics. The technique is not new to climate sciences as well: Torrence and Compo published in 1998 "a practical guide to wavelet analysis" in BAMS. This article is cited 7000 times. Wavelets have also been applied to the analysis of geophysical time series by several authors (Grinsted et al 2004, Ghil 2002,...). This vast literature explains why we did not include any validations of the methodology for toy models.

In the previous version of the paper we gave just a short introduction to the wavelet methods. We admit that, as the reviewer suggests, we could give more precise references on how the technique has been already validated in climate science. This justifies why we don't include any further validation study of the wavelet filters. The new version will contain a more extended review of the relevant wavelet climate-related literature. We want also to remark that the paper is not about wavelet filtering that we take for granted for the reasons specified above. The wavelet filtering is here used to separate coherent and turbulent components. The originality of our analysis lies in analyzing these components separately.

References:

-Torrence, Christopher, and Gilbert P. Compo. "A practical guide to wavelet analysis." Bulletin of the American Meteorological society 79.1 (1998): 61-78.

-Grinsted, Aslak, John C. Moore, and Svetlana Jevrejeva. "Application of the cross wavelet transform and wavelet coherence to geophysical time series."Nonlinear processes in geophysics 11.5/6 (2004): 561-566.

-Ghil, Michael, et al. "Advanced spectral methods for climatic time series."Reviews of geophysics 40.1 (2002).

*REFEREE: "Authors claim that the integral of the ACF detects the predictability. However, for that not only the correlation should be high but the error should be small, which is not shown. "*

**ANSWER**: The link between Correlations decay is well known in dynamical systems, although this result has not been applied so often (or sometimes just implicitly) to climate science. Some supplementary references can be found in:

-Osborne, A. Ro, and A. Provenzale. "Finite correlation dimension for stochastic systems with power-law spectra." Physica D: Nonlinear Phenomena 35.3 (1989): 357-381.

-Govindan, R. B., K. Narayanan, and M. S. Gopinathan. "On the evidence of deterministic chaos in ECG: Surrogate and predictability analysis." Chaos: An Interdisciplinary Journal of Nonlinear Science 8.2 (1998): 495-502.

-Crisanti, A., et al. "Intermittency and predictability in turbulence." Physical review letters 70.2 (1993): 166.

Since this literature is probably unknown in climate science, as the referee is pointing out, we will rewrite the new version of the manuscript better explaining the link between ACF and predictability.

*REFEREE: "Authors test the metrics on one resolution model. However, one needs to show that the wavelet-based separation gives satisfactory results by considering models with different resolutions."*

ANSWER: This is a good suggestion for validating our metrics. We performed the test on higher resolution simulation, namely the medium resolution version of the IPSL model (v5) and compared the results to the low resolution model (v3) analysed

in the previous version of the manuscript. Results are nicely consistent between the two resolutions and we report here some examples that will be included in the new version of the manuscript: Figure 1 shows a comparison of $\Delta\Lambda$ and $\Delta\Upsilon$ between the low resolution (v3 – left panels) and medium resolution (v5 – right panels). The $\Delta$ is computed between $2050-2100$ and $2006-2056$ because the output for v5 are available for this period. The analysis shows that results are consistent and the spatial structures of the indicators are similar.

**REFEREE**: *"Authors claim that the difference between $\Lambda 2055-2105$ and $\Lambda 2005-2055$ detects the predictability. I am wondering about sensitivity of this metric with respect to the time interval."*

**ANSWER**: The reviewer also suggests to perform a sensitivity study with respect to the change in time interval. In the new version of the paper we will show and comment the results for three different time windows:

- 30years [2070 /2100 – 2006/2036] ,

- 40years [2060 /2100 – 2006/2046] ,

- 50years [2050 /2100 – 2006/2056] .

Figure 2 shows $\Delta\Upsilon$ for $u_{700}$ and $\Delta\Lambda$ for $v_{700}$, in the low resolution simulation and for the three different time windows. Coherence among spatial structures is preserved for the variables shown (we will add the analysis for the other cases in the new version of the manuscript) although the intensity of changes is slightly different and generally increases by decreasing the window size. This is expected on the basis of the increased separation in the time periods considered.

Figure 3 summarizes with box-plots the additions requested by the referee. We report results for the two different scenarios, resolutions and time windows (the 30 years

cases are not shown here because the analysis is still running for the medium resolution simulation. We will add it to the new version of the manuscript). It is interesting to notice how the turbulent component $\Delta\Upsilon$ changes with the resolution : we find that adding finer scales corresponds to richer turbulent contributions, as one would expect on theoretical basis.

Overall, we thank the referee for his comments and we believe that these additions increase the range of validity of our results and improve the quality of our work.

*REFEREE: "Moreover, authors need to describe the wavelet-based approach, define what BIC is, and to explain how the parameters were chosen."*

**ANSWER**: We will add these descriptions in the new version of the paper.

The figure panels are labeled:

- RCP 8.5 $u_{700}\Delta\Lambda$ 50years v3
- RCP 8.5 $u_{700}\Delta\Lambda$ 50years v5
- RCP 8.5 $v_{700}\Delta\Lambda$ 50years v3
- RCP 8.5 $v_{700}\Delta\Lambda$ 50years v5
- RCP 8.5 $u_{700}\Delta\Upsilon$ 50years v3
- RCP 8.5 $u_{700}\Delta\Upsilon$ 50years v5
- RCP 8.5 $v_{700}\Delta\Upsilon$ 50years v3
- RCP 8.5 $v_{700}\Delta\Upsilon$ 50years v5

**Fig. 1.** Comparison of $\Delta\Lambda$ and $\Delta Y$ between the low resolution (v3 – left panels) and medium resolution (v5 – right panels). The analysis shows that coherent structures are similar

[Figure]

**Fig. 2.** Comparison of $\Delta Y$ u700 and $\Delta \Lambda$ v700 for three different time windows. Upper panels: 30years [2070 /2100 – 2006/2036]. Central : 40years [2060/2100–2006/2046] . Bottom:50years [2050/2100 – 2006/2056]

[Figure]

**Fig. 3.** Boxplots summarizing the changes among different resolutions (v3 for low resolution and v5 for high resolutions), scenarios and time intervals for each variables and indicators.

---

## Author Response (AR1)

Gif-sur-Yvette,

01/06/2016

Dear Editor,

We have answered to the referees' comments and modified consequently the paper as discussed in the detailed answers. We hope that the paper is now suitable for publication in ESD and we thanks the referees for their interesting comments/remarks which have surely improved the quality of our works.

Best Regards,

Davide Faranda & Dimitri Defrance

*Anonymous Referee #1*

Overall quality. This is a good paper, with some limitations as to the broader conclusions. The authors apply new wavelet based metrics on the changes in circulation under a changing climate and the use of these novel diagnostics is the strongest part of this paper. However, the authors have applied this methodology to a coarse resolution atmospheric simulation that may not be adequate to draw all of the conclusions stated in the paper. This flaw can be rectified, however, by reducing the scope of the conclusions for reasons stated below.

This paper has three main parts: 1) the use of a wavelet filter to separate the coherent and turbulent components of the 700 hPa wind field; 2) The analysis of these fields using two novel metrics in a simulation of the present day climate; 3) the analysis of the changes in these metrics in future climates simulate using RCP 2.5 and RCP 8.5 warming scenarios.

There are questions with respect to the implementation of the first part and the interpretation of the third part. With respect to the implementation of the first part the, filter and separation, the concern is that the turbulent component has a large seasonal cycle contribution; as large as the coherent part of the flow. This appears to be an inadequacy of the filter since one would assume that there should be only a small contribution from seasonal fluctuations in the turbulent component. It would be comforting if the authors at least noted this problem in the paper.

**We discuss this issue in the new version of the manuscript by adding: "The limits of the wavelet filtering technique appear by looking at the spectral peak corresponding to the seasonal cycle which cannot be completely eliminated, although it represents a coherent component of the signal." Our considerations were mostly related to the slope of the spectrum.**

Second, the authors interpret the analysis of their results indicating a relative increase in the coherent component versus the turbulent component under the RCP 8.5 scenario as pointing to an increase in predictability. This may be true, however, only for the scales that are well resolved in the IPSLCM5-LR model used; i.e. structures on the synoptic and planetary scales well resolved by a 3.75°x1.875° mesh. This says nothing about what may happen to meso-scale and convective scale phenomena that are unresolvable with such a coarse resolution as used here.

**The reviewer is right in saying that we have to focus our conclusions on the synoptic and planetary scales resolved by the model. We have understated the results in the new version of the paper. We also includes some new analyses at Medium Resolution (see supplementary material) which add an**

**interesting point to what observed by the referee: increasing the resolution causes also a relative increase of ΔY of u700 and ΔΛ of v700 component (see supplementary material). This means that turbulent contributions become more relevant as finer scales are included in the analysis, although the scale of the analysis still remains much larger than the scales of convective structures.**

Technical corrections :

Pg 2 line 16 ' only change the intensity' should be ' only a change in the intensity'

 Pg 4 line 4 'an higher' should be 'a higher'  Pg 4 line 31 'At the tropics' should be 'In the tropics'

Pg 5 line 6 'ibn' should be 'in' Discussion paper

Pg 5 line 21 'hilited by an alternance' should be 'highlighted by an alteration'

Pg 5 line 29 'of about' should be ' by about'

**The typos will be corrected in the new version of the paper**

*Anonymous Referee #2*

The wavelet-based approach is up to my knowledge a novel methodology for analyzing climate models, though it has been used for image or reservoir reconstructions. I, however, have big doubts that this method is suitable for climatology. Climate modeling demands multi-scale modeling as well but the scale separation is often difficult to de- fine and what is more important there is a multi-scale interaction that evolves in time. Therefore, the method should be first rigorously examined for climate models (starting from toy models and propagating towards more complex models) before drawing the conclusions about the climate system itself.

**The wavelet approach has been devised for analyzing turbulent signals containing non-trivial scale separations. The original paper by Farge (1992) contains applications of wavelet filtering for toy models as well as for turbulent complex systems. This paper has, up to date, ~1500 citations, corresponding to just as many applications in complex fluid mechanics. The technique is not new to climate sciences as well: Torrence and Compo published in 1998 "a practical guide to wavelet analysis" in BAMS. This article is cited 7000 times.  Wavelets have also been applied to the analysis of geophysical time series by several authors (Grinsted et al 2004, Ghil 2002,…). This vast literature explains why we did not include any validations of the methodology for toy models.**

**In the previous version of the paper we gave just a short introduction to the wavelet methods. We admit that, as the reviewer suggests, we could give more precise references on how the technique has been already validated in climate science. This justifies why we did not do any validation study. The new version contains a more extended review of the relevant wavelet climate-related literature.**

**We want also to remark that the paper is not about wavelet filtering that we take for granted for the reasons specified above. The wavelet filtering is here used to separate coherent and turbulent components. The originality of our analysis lies in analyzing these components separately.**

**References (included in the new version of the manuscript)**

**-Torrence, Christopher, and Gilbert P. Compo. "A practical guide to wavelet analysis."** *Bulletin of the American Meteorological society* **79.1 (1998): 61-78.**

-Grinsted, Aslak, John C. Moore, and Svetlana Jevrejeva. "Application of the cross wavelet transform and wavelet coherence to geophysical time series." *Nonlinear processes in geophysics* 11.5/6 (2004): 561-566.

-Ghil, Michael, et al. "Advanced spectral methods for climatic time series." *Reviews of geophysics* 40.1 (2002).

Authors claim that the integral of the ACF detects the predictability. However, for that not only the correlation should be high but the error should be small, which is not shown.

**The link between Correlations decay and predictability is well known in dynamical systems, although this result has not been applied so often (or sometimes just implicitly) to climate science. Some supplementary references can be found in:**

**-Osborne, A. Ro, and A. Provenzale. "Finite correlation dimension for stochastic systems with power-law spectra."** *Physica D: Nonlinear Phenomena* **35.3 (1989): 357-381.**

**-Govindan, R. B., K. Narayanan, and M. S. Gopinathan. "On the evidence of deterministic chaos in ECG: Surrogate and predictability analysis."** *Chaos: An Interdisciplinary Journal of Nonlinear Science* **8.2 (1998): 495-502.**

**-Crisanti, A., et al. "Intermittency and predictability in turbulence."** *Physical review letters* **70.2 (1993): 166.**

**Since this literature is probably unknown in climate science,   we give such references in the new version of the manuscript.**

Authors test the metrics on one resolution model. However, one needs to show that the wavelet-based separation gives satisfactory results by considering models with different resolutions.

**This is a good suggestion for validating our metrics. We performed several tests on higher resolution simulation, namely the medium resolution version of the IPSL model (MR) and compared the results to the low resolution model (LR) analyzed in the previous version of the manuscript. Results are nicely consistent between the two resolutions and we report them in the new supplementary data of the manuscript. The analysis also shows that the spatial structures of the indicators are similar.**

Authors claim that the difference between $\Lambda_{2055-2105}$ and $\Lambda_{2005-2055}$ detects the predictability. I am wondering about sensitivity of this metric with respect to the time interval.

**The reviewer also suggests to perform a sensitivity study with respect to the change in time interval. In the supplementary material of the paper we show and comment the results for three different time windows:**

1) **30years [2070 /2100 – 2006/2036] ,**
2) **40years [2060 /2100 – 2006/2046] ,**
3) **50years [2050 /2100 – 2006/2056] .**

**Coherence among spatial structures is preserved although the intensity of changes is slightly different and generally increases by decreasing the window size. This is expected on the basis of the increased separation in the time periods considered.**

**Figure 7 in the supplementary summarizes with box-plots the addition requested by the referee. We report results for the two different scenarios, resolutions and time windows.  It is interesting to**

**notice how the turbulent component Δγ changes with the resolution. We find that adding finer scales corresponds to richer turbulent contributions, as one would expect on theoretical basis.**

**Overall, we thank the referee for his comments and we believe that these additions increase the range of validity of our results.**

Moreover, authors need to describe the wavelet-based approach, define what BIC is, and to explain how the parameters were chosen.

**We added the BIC formula/explanation in the new version of the paper.**

[revised manuscript text omitted]